# Use of Dexamethasone via Two Methods in PENG Block for Patients Undergoing Femoral Fracture Surgery: A Prospective Observational Study

**DOI:** 10.3390/jcm14228040

**Published:** 2025-11-13

**Authors:** Emine Ozdemir, Ebru Kelsaka, Halil Cebeci

**Affiliations:** 1Alaca State Hospital, 19600 Corum, Turkey; 2Department of Anesthesiology and Reanimation, Faculty of Medicine, Ondokuz Mayıs University, Kurupelit Campus, 55139 Samsun, Turkey; ebruk@omu.edu.tr (E.K.); halilcebeci55@gmail.com (H.C.)

**Keywords:** hip fracture, pericapsular nerve group block, dexamethasone

## Abstract

**Objectives**: This study investigates the effectiveness of dexamethasone when utilized as an adjunct agent in enhancing the outcomes of a pericapsular nerve group (PENG) block compared to its systemic administration for managing pain in patients having surgery for femoral fractures. **Methods**: This study enrolled 44 patients who received a PENG block following spinal anesthesia. Two groups were formed by stratifying the patients, involving those in whom dexamethasone was received through perineural administration (Group P, *n* = 22) and those in whom it was received through systemic administration (Group S, *n* = 22). Information concerning the demographic features of the patients, along with operative and postoperative details, was meticulously documented for analysis. The patients’ pain levels were recorded using the numerical rating scale (NRS) at multiple time points. **Results:** In the perineural dexamethasone group, the length of time of sensory and motor blockade and the time elapsed until the initial analgesic requirement were longer (*p* < 0.001). The consumption of tramadol and the pain scores measured were noted to be decreased. (*p* < 0.001). There were no notable distinctions regarding patient mobilization or the length of inpatient stay. **Conclusions**: The perineural administration of dexamethasone in PENG block procedures may provide more effective analgesia for surgeries involving femoral neck fractures. By minimizing the use of systemic opioids, it may also help mitigate potential side effects. These findings indicate that perineural dexamethasone could serve as a beneficial adjuvant agent for patients undergoing femoral neck fracture surgeries.

## 1. Introduction

Hip fractures are prevalent in society and cause significant mortality and morbidity in individuals of the geriatric age group [1]. The pain experienced by patients during the perioperative period may cause some metabolic and endocrine changes in the body, thus causing the pain to become chronic [2]. Non-steroidal anti-inflammatory drugs and opioids, which are commonly prescribed for pain relief in the geriatric population, may have adverse effects on respiratory, hepatic, and renal functions. Regional anesthesia techniques may be employed throughout the perioperative phase to mitigate the side effects of these medications [3]. Utilizing regional anesthesia techniques has been associated with reduced opioid consumption, shortened hospital stays, and increased patient satisfaction [4].

Traditional blocks can cause motor blockade and are limited in efficacy, as they do not target all sensory nerves involved in the hip joint. Pericapsular nerve group (PENG) block is a fascial area block that provides optimal analgesia while preserving motor function [5]. The PENG block offers analgesia by selectively targeting the femoral, obturator, and accessory obturator nerves. The anterior capsule of the hip joint receives its sensory innervation through these nerves and importantly, this technique achieves pain relief while preserving motor function. This improves patient’s mood, sleep, appetite, and general well-being [6].

One of the primary drawbacks of regional anesthesia is severe pain experienced after the local anesthetic effect has passed. Therefore, adjuvant agents such as epinephrine, bicarbonate, dexamethasone, and clonidine can be used to increase efficacy of local anesthetics in nerve blocks [7]. In PENG block applications, it is recommended to add dexamethasone as an adjuvant to extend the effect of the local anesthetic and enhance the overall quality of the block [8,9,10].

Although the mechanism of action is not fully explained when dexamethasone was administered perineurally, it could prolong the duration of peripheral nerve blocks by an average of 6 h compared to the placebo [10]. This phenomenon could potentially be associated with the systemic absorption of dexamethasone, its modulatory effect on C fibers and its local vasoconstrictor properties [7].

The principal goal of this study was to explore the impact of perineural and systemic dexamethasone administration on pain scores and potential complications within the initial 24 h following PENG block for analgesia in individuals undergoing surgery for femoral fracture.

## 2. Materials and Methods

### 2.1. Trial Design

This single-center, prospective observational study took place at a tertiary health center’s Department of Anesthesiology and Reanimation between August 2023 and October 2023. Patients were consecutively enrolled according to their eligibility during the study period. The decision on whether dexamethasone would be administered perineurally or systemically was made by the attending anesthesiologist based on clinical practice preferences, not by random allocation. Therefore, no random sequence generation or allocation concealment was performed. The study was conducted as a prospective observational design, and the groups were formed naturally without any intervention from the researchers regarding assignment. Although group sizes were equal, this was coincidental due to the total number of eligible patients within the study period. Before enrolling in the study, all participants provided written informed consent. The research report was written according to the Strengthening the Reporting of Observational Studies in Epidemiology (STROBE) guidelines. Ethical approval was obtained from the Ondokuz Mayıs University Ethics Committee (Approval No: 2023/239, Approval Date: 23 August 2023) and adhered to the ethical principles outlined in the Declaration of Helsinki.

### 2.2. Participants

Patients aged between 50 years and 85 years with a body mass index below 35 kg/m^2^ and categorized as American Society of Anesthesiologists (ASA) classes II-IV, who underwent surgery for a femoral fracture under spinal anesthesia, qualified for participation in the study. Exclusion criteria encompassed a history of alcohol or drug addiction, opioid use for longer than 4 weeks, chronic pain disorders, allergies to local anesthetics and opioids, contraindications for regional anesthesia procedures, severe psychiatric disorders, such as dementia, that limit patient cooperation, or if the patients were assessed as unsuccessful in the dermatome examination after the block application. As this was an observational study conducted under real clinical conditions, neither patients nor outcome assessors were blinded to group allocation, which could potentially introduce bias in subjective outcome assessments.

The sample size was estimated using G*Power 3.1 for a one-tailed *t*-test comparing two independent means. An effect size of Cohen’s *d* = 0.90, α = 0.05, and power (1−β) = 0.90 were assumed, resulting in 22 patients per group (44 total). The effect size was based on the study by Allard et al. which reported a similar standardized difference in postoperative pain outcomes between regional block techniques [11].

Finally, the study recruited 44 patients, among whom 22 received perineural administration of dexamethasone, and 22 received systemic administration of dexamethasone (Figure 1). No missing data were encountered, and complete case analysis was performed.

### 2.3. Interventions

#### 2.3.1. Spinal Anesthesia Management

All patients underwent ASA standard monitoring, which includes electrocardiography, non-invasive blood pressure measurement, and peripheral oxygen saturation monitoring. A 20-G intravenous cannula was utilized to initiate an infusion of 2–3 mL/kg of Ringer’s lactate solution.

Spinal anesthesia was chosen as the anesthesia method. Following antiseptic protocols, patients in a sitting position received an intrathecal injection at the L3-4 interspace using a 25-G Quincke-tipped needle (Egemen^®^ Izmir, Turkey). After observing free cerebrospinal fluid flow from the spinal needle, spinal anesthesia was induced with a dose 12 mg of 0.5% heavy bupivacaine and 10 µg of fentanyl. Intrathecal fentanyl (10 µg) was added to 12 mg of 0.5% heavy bupivacaine for all patients as per institutional protocol to enhance intraoperative comfort and minimize pain during surgical positioning. This same regimen was applied to both groups to avoid variability related to intraoperative analgesia. No additional sedatives or intraoperative analgesics were administered other than those described in the study protocol.

#### 2.3.2. Technique of PENG Block

Following the administration of spinal anesthesia, patients were positioned supine, and a low-frequency curvilinear ultrasound probe (3–16 MHz, GE Healthcare LOGIQ E R7) was placed parallel to the inguinal ligament. The probe was rotated 45° and aligned for clear visualization of the spina iliaca anterior inferior, iliopubic eminence, and femoral artery. Using an in-plane technique, a 22-G 80 mm hyperechoic needle (Stimupleks^®^ Ultra 360^®^, 80 mm, Braun, Germany) was inserted between the psoas muscle tendon and iliopubic eminence. Following hydrodissection, a mixture of local anesthetic drugs was injected. The localization of the medication within the area spanning from the psoas muscle tendon to the iliopubic eminence was observed.

Group P (Perineural): Following hydrodissection (2 mL saline), 20 cc of 0.25% bupivacaine (Buvasin, Vem, Turkey) and 4 mg of dexamethasone (Deksamet, Osel, Turkey) were injected into the fascial area, and 1 cc of saline was administered intravenously.

Group S (Systemic): Following hydrodissection (2 mL saline), 20 cc of 0.25% bupivacaine (Buvasin, Vem, Turkey) and 1 cc of saline were injected into the fascial area, and 4 mg of dexamethasone (Deksamet, Osel, Turkey) was administered intravenously.

The choice of 4 mg preservative-free dexamethasone was based on randomized dose–response studies indicating a ceiling effect near 4 mg for prolonging the duration of peripheral nerve block analgesia without added benefit from higher doses [8,9]. This dose has been widely used in clinical trials as an adjuvant to long-acting local anesthetics, including in emerging studies of PENG and hip surgery blocks. Perineural dexamethasone was used in a preservative-free formulation, and all administrations were performed under ultrasound guidance. Current systematic reviews and Cochrane analyses report no increase in neurologic complications with perineural use compared with placebo or intravenous routes, while noting that perineural steroid use remains off-label and should be applied cautiously in patients with pre-existing neuropathy [7].

The PENG block was performed immediately after spinal anesthesia, before the surgical incision. Sensory level assessment of the PENG block was conducted during spinal anesthesia; therefore, precise dermatomal mapping specific to the PENG distribution could not be evaluated. To minimize the confounding effect of spinal anesthesia, all postoperative pain assessments and block-related parameters (NRS scores, duration of sensory and motor block, and time to first rescue analgesia) were recorded only after the regression of spinal anesthesia (at 6 h, 12 h, and 24 h postoperatively).

#### 2.3.3. Assessment of Pain and Analgesia Protocol

The patient’s pain levels were quantified at various intervals using the Numerical Rating Scale (NRS: before spinal anesthesia, in recovery unit, at rest between 6 and 12 h postoperatively, and both at rest and during activity (15° passive leg raise) at 24 h postoperatively. Postoperative mobilization time was defined as the interval from the end of surgery to the first assisted ambulation (walking with support using a walker or crutches). Bedside sitting was encouraged earlier but not recorded as a time variable.

All patients routinely received 50 mg of intravenous dexketoprofen every 8 h (maximum daily dose of 150 mg) and 1 g of intravenous acetaminophen every 6–8 h (maximum daily dose of 4 g) for 24 h postoperatively.

Regarding patients with an NRS score of ≥4 during the postoperative follow-up period, rescue analgesia was provided with 0.5–1 mg/kg of intravenous tramadol every 4–6 h (maximum daily dose of 400 mg). Total opioid consumption over 24 h was converted to oral morphine equivalents (OME) using the conversion factor 10:1 (tramadol–morphine). The time at which tramadol was first requested was recorded as the “the time elapsed until the initial analgesic requirement”. If the NRS score remained ≥4 after tramadol administration, 0.05 mg/kg of intravenous morphine was administered.

#### 2.3.4. Hemodynamic Monitoring and Rescue Protocol

Hemodynamic events were defined a priori as follows:

Hypotension = SBP < 90 mmHg or MAP < 65 mmHg or ≥20% decrease from baseline; hypertension = SBP > 160 mmHg or ≥20% increase from baseline; bradycardia = HR < 50 bpm; tachycardia = HR > 100 bpm.

Rescue protocol:

Hypotension was treated with norepinephrine 4–8 mcg iv boluses (infusion 0.25–1 µg/kg/min if needed). Bradycardia was treated with atropine 0.5 mg IV (repeat to 3 mg max). Hypertension was treated with gliseril trinitrate 50–100 mcg iv boluses per hemodynamic profile. Tachycardia was managed with esmolol 10–20 mg iv boluses as clinically indicated. All interventions were recorded with dose and timing.

#### 2.3.5. Blood Glucose Levels Monitoring

Capillary blood glucose levels were measured at two standardized time points: preoperatively (fasting, before anesthesia induction) and 6 h postoperatively during routine laboratory checks. The perioperative administration of dexamethasone (4 mg) was performed immediately after spinal anesthesia and before the surgical incision in both groups. A clinically significant change in blood glucose was defined as an increase ≥50 mg/dL from baseline or any postoperative value ≥180 mg/dL, in accordance with previous perioperative glycemic studies.

#### 2.3.6. PONV Assessment

Postoperative nausea and vomiting (PONV) were assessed using a validated 4-point scale (0 = none; 1 = nausea; 2 = severe nausea; 3 = vomiting) at 0–6 h (early) and 6–24 h (late) windows. An amount of 4 mg ondansetron was administered iv as rescue antiemetic.

### 2.4. Outcomes

The primary outcome of this study was defined as the time to first rescue analgesia following surgery. Secondary outcomes included postoperative NRS scores at multiple time points (6 h, 12 h, and 24 h at rest and activity), duration of sensory and motor block, and total tramadol consumption.

### 2.5. Statistical Analysis

Data from comparable studies were employed to perform a statistical power analysis to determine the sample size for the study. With an effect size of d = 0.90 and an alpha error of 5%, the required sample size for 95% confidence interval (CI) and 90% study power was calculated to be 44 patients, with 22 in both groups. Statistical analysis was performed utilizing the SPSS version 23.0 software (IBM Corp., Armonk, NY, USA), Jamovi version 2.3.28, and JASP version 0.17.3. Continuous variables are expressed as mean ± standard deviation (SD) for normally distributed data or as median [interquartile range, IQR] for non-normally distributed data, whereas categorical variables are described in terms of numbers and frequencies. The normality distribution of the numerical data was evaluated by Shapiro–Wilk, Kolmogorov–Smirnov, and Anderson–Darling tests. The independent samples *t*-test was used to compare two independent groups for variables with a normal distribution, while the Mann–Whitney U test was used for non-normally distributed numerical variables. For statistical comparisons of measurements taken over time, repeated measures analysis of variance (ANOVA) was utilized for numerical variables that were normally distributed. Additionally, an analysis of covariance (ANCOVA) was conducted to control for baseline pain scores (pre-spinal NRS) as a covariate when comparing postoperative NRS values between groups. This adjustment was applied to minimize the potential confounding effect of baseline pain differences. Conversely, the numerical variables that deviated from a normal distribution were analyzed using the non-parametric Friedman test. A *p*-value below 0.05 was viewed as statistically significant.

## 3. Results

The total participant count in the study was 44, evenly divided with 22 patients in each group. The analysis showed that there were no statistically significant variations in demographic characteristics between the two groups. There were no observed differences between both groups regarding the amount of bleeding, transfusion volume, or surgical duration (*p* > 0.05). Both groups of patients were mobilized at 24 h, and the median hospitalization period was 4 days (Table 1).

The time to reach the T10 sensory block level was 4 [3–5] min in Group P and 5 [5–6] min in Group S (*p* < 0.001). The time to achieve a Bromage score of 3 was 2 min longer in Group S than in Group P; however, this difference did not attain statistical significance (*p* = 0.2). The median duration of sensory block was 232 [200–250] min in Group P and 180 [160–185] min in Group S (*p* < 0.001). The median duration of motor blocks was recorded as 175 [150–180] and 140 [120–150] min in Groups P and S, respectively (*p* < 0.001) (Table 2).

The sensory levels in patients were monitored at intervals of 1, 5, 10, 20, 45, and 60 min. In both groups, the time required to attain the T10 sensory block level was 5 min in Group P and 10 min in Group S (Table 3). In Group P, sensory block reached the T6 level in all patients within 20 min and was sustained at this level for approximately 60 min. However, in Group S, the sensory block only reached the T6 level in nine patients at 20 min. Nonetheless, by 60 min, the sensory block regressed to the T10 level in 50% of the cases. Notably, patients who received perineural dexamethasone achieved an earlier onset of sensory block reaching the T6 level and had a prolonged duration of action at this level. (Table 3). Both groups exhibited similar motor block degrees throughout these time intervals; therefore, no statistically significant differences were identified in motor block grades (*p* = 0.18).

The time elapsed until the initial analgesic requirement was 540 [480–600] min in Group P and 300 [250–360] min in Group S (*p* < 0.001). The tramadol doses required for rescue analgesic were 45 [0–50] mg in Group P and 70 [50–100] mg in Group S (*p* < 0.001). Total 24 h opioid consumption (oral morphine equivalents, mg) was 4.5 mg in Group P and 7 mg in Group S (mean difference 2.5 mg; %95 CI). Therefore, the time elapsed until the initial analgesic requirement was longer in the group receiving perineural dexamethasone and the total tramadol dose administered for rescue analgesia was lower compared to the systemic dexamethasone group (Table 4).

The primary outcome of this study was the time to first rescue analgesia, which was significantly longer in the perineural dexamethasone group compared with the systemic group (*p* < 0.001). Other parameters such as postoperative NRS scores at different time points, duration of sensory and motor block, and total tramadol consumption were analyzed as secondary outcomes and demonstrated consistent results in favor of the perineural group.

The differences in NRS scores before spinal anesthesia were statistically significant between patients receiving perineural dexamethasone and those receiving systemic dexamethasone (*p* = 0.036). Multiple times after surgery, Group P showed significantly lower NRS scores during both rest and activity periods (*p* < 0.001) (Table 4).

After adjusting for baseline NRS using ANCOVA, postoperative pain scores at 6 h, 12 h, and 24 h (both at rest and during activity) remained significantly lower in the perineural dexamethasone group compared with the systemic group (*p* < 0.001) (Table 4).

A linear mixed-effects model including fixed effects for group, time, and their interaction (group × time), and a random intercept for each subject was applied to analyze repeated NRS measurements. Baseline NRS (before spinal anesthesia) was included as a covariate. The analysis showed significant main effects of time (*p* < 0.001) and group (*p* = 0.014), as well as a significant group × time interaction (*p* = 0.009). Estimated marginal means (adjusted for baseline NRS) and 95% CIs are presented (Figure 2).

During the intraoperative period, six patients in Group P experienced hypotension and two patients experienced hypertension. Conversely, in Group S, four patients experienced hypotension, and three patients experienced hypertension. Bradycardia was observed in one patient in both Groups P and S. Patients with hypotension required norepinephrine infusion. Hypertensive episodes were treated with boluses of gliseril trinitrate. In Group P, one patient with bradycardia received a bolus of 0.5 mg atropine. The incidence of complications was similar across both groups. (*p* > 0.05).

In post-anesthesia care unit (PACU), the tremor score was determined as “0”. The incidence of nausea and vomiting was comparable between the two groups with no statistically significant variation detected (*p* > 0.05). At 6 h postoperatively, the change in blood glucose level was calculated as 38 mg/dL in Group P and 40 mg/dL in Group S (*p* = 0.4) (Table 5). None of the patients exhibited a clinically significant rise (≥50 mg/dL) or postoperative hyperglycemia ≥180 mg/dL.

No patients experienced paresthesia during needle advancement, vascular puncture, hematoma, local anesthetic systemic toxicity, or block-site infection. Additionally, no new or persistent motor deficits were observed at 24–48 h postoperatively.

## 4. Discussion

In this study, perineural administration of dexamethasone in the PENG block applied to patients receiving surgical treatment for femoral fractures prolonged sensory and motor block durations, reduced pain scores, extended time elapsed until the initial analgesic requirement, and decreased opioid use compared to systemic administration.

In the literature review, we found no studies evaluating the influence of spinal anesthesia on the time span of the sensory and motor blockade in fascial plane blocks. However, there are studies evaluating the influence of dexamethasone on the time span of sensory and motor blockade in plexus blocks Godbole et al. performed supraclavicular block using a nerve stimulator in 60 patients undergoing forearm surgery [12,13]. Patients were randomly assigned to two groups: one group received perineural administration of 20 mL of 0.5% bupivacaine and 0.05 mg/kg dexamethasone, while the other group received intravenous administration of 0.05 mg/kg dexamethasone after a block composed of 20 mL of 0.5% bupivacaine. The commencement of sensory and motor blocks was similar in both groups, with no significant discrepancies noted. The time span of sensory block was 876 min in the group where dexamethasone was administered perineurally and 560 min in the group where it was administered intravenously. In the perineural dexamethasone group, the motor block lasted 326 min longer than in the systematically group.

In the current study, the sensory block duration was recorded as 232 min in the group where dexamethasone was administered perineurally and 180 min in the group where it was administered systemically. Additionally, the motor block duration was found to be 35 min longer in the perineural group compared to the systemic administration group. This difference may be attributed to the local effect of dexamethasone and local anesthetic on nociceptors and glucocorticoid receptors in the fascial space, in addition to the systemic effect. Furthermore, in our study, the sensory block reached the T6 level more rapidly and lasted longer in the group receiving perineural dexamethasone. This finding indicates that dexamethasone applied to the fascial plane may affect the sensory block level under spinal anesthesia and thus prolong the block duration. We believe that this finding is clinically valuable. The early onset and extended duration of sensory block provide significant benefits in initiating surgery promptly and managing prolonged surgeries under spinal anesthesia. Moreover, by delaying the time elapsed until the initial analgesic requirement postoperatively, it contributes to a reduction in opioid consumption.

When dexamethasone is incorporated as a supplement to local anesthetics in fascial plane blocks, the length of analgesic effect is prolonged, and pain scores are reduced compared to systemic dexamethasone administration [14,15,16]. Gupta et al. investigated 90 patients undergoing cesarean section who received a transversus abdominis plane (TAP) block [15]. The patients were assigned to two groups: the first group received a block composed of 25 mL of 0.375% ropivacaine, and the second group received a block composed of 25 mL of 0.375% ropivacaine and 4 mg of dexamethasone. According to the study findings, the group receiving dexamethasone exhibited lower pain scores, showing a decreased need for additional analgesic and delayed onset of the first analgesic requirement. In another study including patients undergoing cesarean section, 87 patients who underwent bilateral TAP blocks, divided into three groups, were examined [16]. The first group received a block composed of 40 mL of 0.25% bupivacaine combined with 8 mg of perineural dexamethasone. The second group received a block composed of 40 mL of 0.25% bupivacaine combined 8 mg of intravenous dexamethasone, and the third group received a block composed of 40 mL of 0.25% bupivacaine only. The time elapsed until the initial analgesic requirement in the perineural dexamethasone group was found to be 1.5 h longer compared to the intravenous administration group and 4 h longer compared to the third group. In research performed by Singariya et al., bilateral rectus sheath blocks in patients scheduled for midline laparotomies were performed [14]. One group received a block composed of 20 mL of 0.25% levobupivacaine and 8 mg of dexamethasone, while the other group received a block composed of 20 mL of 0.25% levobupivacaine and 8 mg of intravenous dexamethasone. Their findings indicated that perineural dexamethasone extended analgesia by 200 min over systemic administration and led to lower opioid usage and pain levels.

In this study, in line with the current literature, we identified that dexamethasone administered to the fascial plane prolonged the time elapsed until the initial analgesic requirement by 240 min compared to systemic administration and reduced the tramadol dose used for rescue analgesia by 25 mg. According to the opioid equianalgesic dose calculator based on the national validity of GlobalRPH, this dose was found to be equivalent to 2.5 mg of morphine. Despite the fact that this variation reached statistical significance, it was not clinically relevant. We posit that this variance stems from the regulation of C fibers and vasoconstriction via local glucocorticoid receptors, leading to a decline in systemic absorption of the local anesthetic.

In this study, consistent with the literature, NRS scores in patients who received perineural dexamethasone were found to be significantly lower at 6–12 and 24 h postoperatively during rest and activity compared to preoperative values [15,16]. In contrast, NRS scores were higher in patients in the systemic dexamethasone group before spinal anesthesia. Although a significant baseline difference in pre-spinal NRS was observed between the groups, the ANCOVA-adjusted analysis confirmed that postoperative pain reduction in the perineural dexamethasone group persisted after controlling for baseline pain. Therefore, the analgesic benefit observed was unlikely due solely to initial pain differences. In addition to baseline NRS, age, and sex, the type of surgical procedure (partial hip replacement vs. proximal femoral nail fixation) was included as a covariate in the ANCOVA model to control for potential confounding related to surgical approach and tissue trauma. After adjustment, the group effect on postoperative pain scores remained statistically significant (*p* = 0.018), indicating that the analgesic benefit of perineural dexamethasone was independent of the type of surgery performed.

Furthermore, Desmet et al. and McHardy et al. found that mean postoperative blood glucose levels were notably elevated in the group receiving intravenous dexamethasone compared to the perineural dexamethasone group; however, the difference was not clinically significant [17,18]. In the study conducted by Desmet et al. the blood glucose level showed an average increase of 5.5 mg/dL in the intravenous group and 3.6 mg/dL in the perineural group [17]. McHardy et al. reported an average difference of 6.2 mg/dL in the intravenous group versus the perineural group, whereas Chun et al. detected no significant difference between them [18,19]. In our study, blood glucose monitoring followed a standardized protocol. Despite the known transient hyperglycemic effect of corticosteroids, neither group demonstrated clinically significant elevations in perioperative glucose values. The median postoperative increase remained below the predefined threshold of ≥50 mg/dL. These findings align with previous reports indicating that single-dose perioperative dexamethasone, whether perineural or systemic, exerts minimal and clinically insignificant glycemic impact in surgical patients.

Additionally, no difference in nausea and vomiting was recorded between both groups in our study. These findings indicate that dexamethasone administered in the PENG block may have both local and systemic effects.

The 4 mg dose of dexamethasone used perineurally was selected based on dose-finding evidence showing maximal analgesic prolongation at or below 4 mg [9]. Clinical trials and meta-analyses support the safety of preservative-free perineural dexamethasone, with no increased risk of neurotoxicity compared with intravenous administration [10,20]. Nevertheless, perineural steroid administration is off-label, and potential adverse effects should be considered [21]. In this study, no block-related complications or motor deficits at 24–48 h were identified.

## 5. Limitations

First, since the PENG block was administered to patients after spinal anesthesia, it was not possible to assess the onset time of sensory block, sensory dermatome levels, and quadriceps muscle strength specific to the PENG block. However, all postoperative outcomes were assessed after the regression of spinal anesthesia, reducing the likelihood of confounding in analgesic comparisons. Although intrathecal fentanyl and standardized multimodal postoperative analgesia were used in all patients to ensure consistency, these concomitant interventions might have attenuated the magnitude of between-group differences attributable solely to the dexamethasone administration route. Furthermore, as this was an observational study without randomization or blinding, the potential for selection, performance, and measurement bias cannot be entirely excluded. Another limitation of this study is the absence of blinding for both patients and outcome assessors. Because of the non-randomized observational design, residual confounding and selection bias cannot be fully excluded. Therefore, these findings should be interpreted as hypothesis-generating. Because of the small sample size, a reliable multivariable regression analysis including demographic and clinical covariates (ASA, sex, procedure type, and comorbidities) could not be performed. Future randomized studies with larger sample sizes are needed to validate these associations using regression modeling.

## 6. Conclusions

In conclusion, adding adjuvant dexamethasone to the PENG block extends both sensory and motor block durations, thereby enhancing analgesic efficacy in hip surgeries. In this prospective observational study, perineural dexamethasone was associated with longer analgesic duration compared with systemic administration following PENG block in elderly patients. However, due to the non-randomized design, limited sample size, and potential baseline differences, these results should be interpreted as hypothesis-generating. Further randomized, double-blinded controlled trials are warranted to confirm these findings.

## Figures and Tables

**Figure 1 jcm-14-08040-f001:**
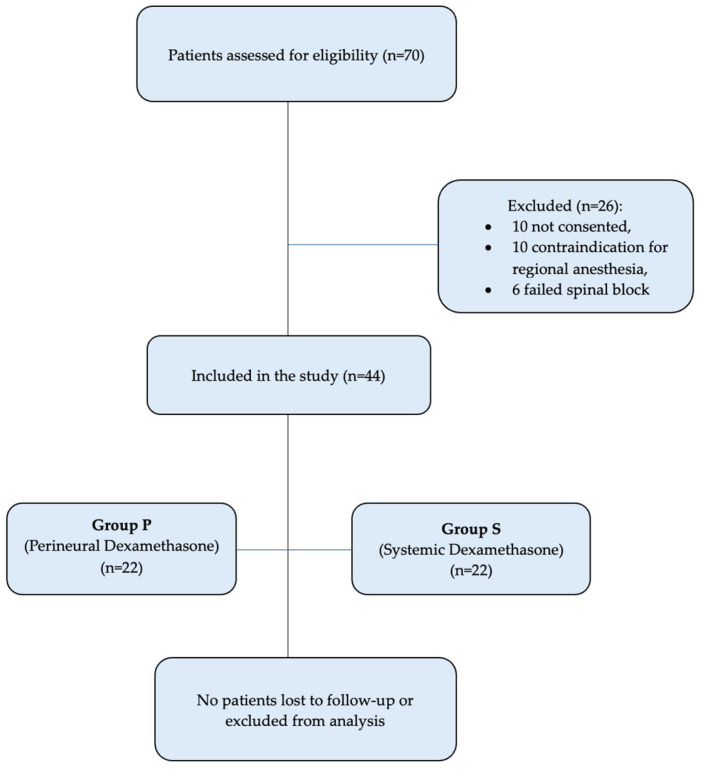
STROBE-compliant flow diagram showing patient screening, exclusions with reasons, group allocation, and follow-up status.

**Figure 2 jcm-14-08040-f002:**
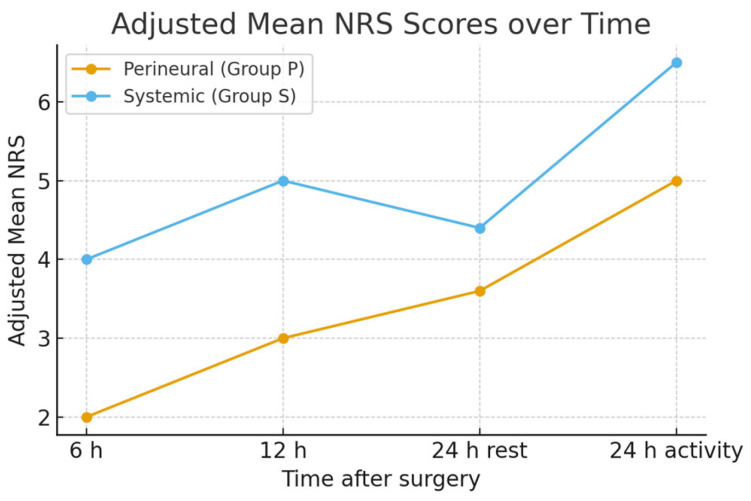
Adjusted mean NRS.

**Table 1 jcm-14-08040-t001:** Demographic and surgical information of patients.

	Group P (*n* = 22)	Group S (*n* = 22)	*p*-Value
Age	77.8 ± 7.9	77.2 ± 7.7	0.804 ^t^
Gender			
Female	14 (63.6%)	15 (68.2%)	0.999 ^X2^
Male	8 (36.4%)	7 (31.8%)	
Height (cm)	161.4 ± 6.9	162.2 ± 7.7	0.696 ^t^
Weight (kg)	73.2 ± 14.1	74.4 ± 15.8	0.803 ^t^
ASA Score			
I	0 (0.0)	0 (0.0)	0.549 ^t^
II	3 (13.6%)	1 (4.5%)	
III	16 (72.7%)	16 (72.7%)	
IV	3 (13.6%)	5 (22.7%)	
Comorbidities			
HT	22 (100.0%)	22 (100.0%)	--
CAD	22 (100.0%)	22 (100.0%)	--
CVD	17 (77.3%)	17 (77.3%)	0.999 ^t^
DM	8 (36.4%)	10 (45.5%)	0.759 ^t^
CRF	1 (4.5%)	1 (4.5%)	0.999 ^t^
Surgery type			
Partial hip replacement	22 (100.0%)	20 (90.9%)	0.488 ^X2^
PFN	0 (0.0%)	2 (9.1%)	
Bleeding (mL)	300 [300–400]	325 [287.5–400]	0.799 ^m^
Crystalloids (mL)	1025 [1000–1500]	1500 [950–1500]	0.615 ^m^
Colloids (mL)	500 [450–500]	500 [462.5–500]	0.596 ^m^
ES (mL)	250 [0–252.5]	250 [0–245]	0.516 ^m^
FFP (mL)	220 [0–220]	200 [0–200]	0.202 ^m^
Operation duration (min)	100 [90–112.5]	90 [90–120]	0.697 ^m^
Mobilization duration (hour)	24 [20–24]	24 [22–24]	0.206 ^m^
LOS (day)	4 [3–5]	4 [4–5]	0.656 ^m^

^t^: Independent samples *t*-test. ^X2^: Pearson chi-square or Fisher’s exact test, ^m^: Mann–Whitney U test. Data are presented as mean ± SD or median [Q1–Q3], number of patients (*n*), and percentage (%). ASA:American Society of Anesthesiologists; CAD: coronary artery disease; CVD: cerebrovascular disease; DM: diabetes mellitus; CRF: chronic renal failure; ES: erythrocyte suspension; FFP: fresh frozen plasma; HT: hypertension; LOS: length of hospital stay; PFN: proximal femoral nail.

**Table 2 jcm-14-08040-t002:** Sensory and motor block durations according to patient groups.

	Group P (*n* = 22)	Group S (*n* = 22)	*p*-Value
Time to reach T10 sensory block (min)	4 [3–5]	5 [5–6]	<0.001 *
Time to achieve Bromage 3 (min)	8 [8–10]	10 [8–10]	0.248
Duration of sensory block (min)	232 [200–250]	180 [160–185]	<0.001 *
Duration of motor block (min)	175 [150–180]	140 [120–150]	<0.001 *

Mann–Whitney U test. Data are presented as mean ± SD or median [Q1–Q3], number of patients (*n*), and percentage (%), * statistically significant difference.

**Table 3 jcm-14-08040-t003:** Levels of sensory block according to patient groups.

		Group P (*n* = 22)	Group S (*n* = 22)	*p*-Value
Sensory block level at 5 min	L1	0	10 (45.5%)	0.001 *
	T10	22 (100%)	12 (54.5%)	
Sensory block level at 10 min	T10	9 (40.9%)	22 (100%)	<0.001 *
	T6	13 (59.1%)	0	
Sensory block level at 20 min	T10	0	13 (59.1%)	<0.001 *
	T6	22 (100%)	9 (40.9%)	
Sensory block level at 60 min	T10	1 (4.5%)	11 (50%)	0.002 *
	T6	21 (95.5%)	11 (50%)	

Pearson chi-square or Fisher–Freeman–Halton test, * statistically significant difference.

**Table 4 jcm-14-08040-t004:** Comparison of pain levels and analgesic requirement between groups.

	Group P (*n* = 22)	Group S (*n* = 22)	*p*-Value
The time elapsed until the initial analgesic requirement (min)	540 [480–600]	300 [250–360]	<0.001 *
Rescue analgesic (mg)	45 [0–50]	70 [50–100]	<0.001 *
NRS Score			
Before spinal block	9.3	9.8	0.036 *
After surgery 6 h	2	4	<0.001 *
After surgery 12 h	3	5	<0.001 *
After surgery 24 h rest	3.6	4.4	<0.001 *
After surgery 24 h activity	5	6.5	<0.001 *

ANCOVA. Data are presented as mean ± SD or median [Q1–Q3], number of patients (*n*) * statistically significant difference. NRS: Numeric rating scale.

**Table 5 jcm-14-08040-t005:** Intraoperative complications and postoperative clinical data of patients.

	Group P (*n* = 22)	Group S (*n* = 22)	*p*-Value
Hypotension	6 (27.3%)	4 (18.2%)	0.719 ^X2^
Hypertension	2 (9.1%)	3 (13.6%)	0.999 ^X2^
Bradycardia	1 (4.5%)	0 (0.0)	0.999 ^X2^
Tachycardia	1 (4.5%)	0 (0.0)	0.999 ^X2^
Tremor score-PACU	0.0 ± 0.0	0.0 ± 0.0	--
Nausea/vomiting	3 (13.6%)	2 (9.1%)	0.999 ^X2^
Amount of change in blood glucose level (mg/dL)	38 [30–40]	40 [30–45]	0.486 ^m^

^X2^: Pearson chi-square or Fisher’s exact test, ^m^: Mann–Whitney U test. Data are presented as mean ± SD or median [Q1–Q3], number of patients (*n*), and percentage (%). PACU: post-anesthesia care unit.

## Data Availability

The data that form the basis of this study’s findings can be accessed through a request to the corresponding author The data are not publicly accessible due to privacy and ethical constraints.

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
