# Peer review of "Use of Dexamethasone via Two Methods in PENG Block for Patients Undergoing Femoral Fracture Surgery: A Prospective Observational Study"

_jcm, 2025, doi:10.3390/jcm14228040_

Round 1

Reviewer 1 Report

Comments and Suggestions for Authors

General aspects

It is a single-center prospective observational study, aiming to compare perineural versus systemic dexamethasone as adjuncts to PENG block for analgesia in femoral-fracture surgery under spinal anesthesia. Forty-four geriatric patients were allocated 1:1 to perineural or intravenous dexamethasone. The perineural group showed longer sensory/motor block durations, delayed time to first rescue analgesia, lower opioid use, and lower NRS scores at several time points, with similar short-term adverse-event profiles and glycemic changes. The topic is clinically relevant; however, important design, analysis, and reporting issues limit confidence in the conclusions.

Major comments

Study design, allocation, and risk of bias

The manuscript describes a “prospective observational” design but presents equal group sizes and protocolized interventions as if randomized. Clarify how patients were assigned (sequence generation, concealment). Without randomization and blinding, baseline imbalances and performance/measurement bias are likely.

Baseline pain (pre-spinal) differs significantly between groups (p=0.036), suggesting allocation bias that could influence subsequent analgesic outcomes. Adjust analyses for baseline NRS and other covariates.

Outcome assessors and patients appear unblinded. Given subjective endpoints (NRS, rescue timing), lack of blinding materially increases bias.

Concomitant anesthesia/analgesia as confounders

All patients received spinal anesthesia with intrathecal fentanyl plus multimodal postoperative analgesia. These co-interventions may attenuate between-group differences and complicate attribution to dexamethasone route. Please justify intrathecal fentanyl use and standardize/report intraoperative sedatives/analgesics.

Surgical procedures were mostly partial hip replacements, but two PFNs occurred in Group S only. Surgical approach and tissue trauma differ and can affect pain trajectories; perform sensitivity analyses excluding PFN cases or adjust for procedure type.

Primary endpoint and statistical approach

A single, pre-specified primary endpoint is not clearly defined. The manuscript reports multiple time-point NRS measurements and several secondary outcomes without control for multiplicity. Define one primary outcome (e.g., time to first rescue analgesia) and specify hierarchical testing or adjust p-values.

Repeated measures (NRS at multiple times) are more appropriately analysed with mixed-effects models (group, time, group×time), including baseline NRS as a covariate, rather than multiple Mann–Whitney tests. Please re-analyze accordingly and present effect sizes with 95% CIs.

The power calculation (d=0.90) is not referenced and appears based on a large assumed effect. Powering an observational comparison on such an effect risks Type I error inflation with multiple outcomes. Provide the source study and justify the effect size and chosen primary endpoint.

Measurement of block characteristics under spinal anesthesia

PENG performance and sensory level assessments were done after spinal anesthesia, compromising meaningful dermatomal mapping to the block and potentially conflating spinal with PENG effects. This is acknowledged as a limitation, but it materially affects internal validity. Consider focusing endpoints on postoperative time windows after spinal regression and explicitly document timing relative to spinal offset.

Dosing rationale and safety

The choice of 4 mg dexamethasone (perineural vs IV) requires a pharmacological rationale and citations specific to hip blocks/PENG. Discuss evidence on perineural steroid safety (neurotoxicity risk, off-label status) and report any block-related complications (paresthesia, hematoma, LAST, infection) and 24–48 h motor function.

Glycemic outcomes: define measurement schedule (fasting vs random; perioperative steroid timing), thresholds for clinically significant change, and prespecified subgroups (e.g., diabetes). Current reporting (“change in blood glucose level”) without standardized timing limits interpretability.

Adverse events: definitions and management

Provide operational definitions for hypotension, hypertension, bradycardia, and tachycardia (numerical thresholds), and detail vasoactives dosing. Report PONV with a validated scale and time windows. A CONSORT-style harms table would help.

Data integrity and table issues

Table 1 lists 100% prevalence for HT and CAD in both groups (implausible and likely a data/labeling error). Also, “CVD” may overlap with CAD; define each comorbidity clearly.

Units and formats are inconsistent (e.g., decimal commas, “median ± SD”). Use median [IQR] for non-normal variables, mean ± SD for normal, and ensure consistency across text and tables.

The sensory level table implies thoracic dermatomes under spinal anesthesia; ensure assessments reflect PENG effects, not spinal spread.

Flow diagram (Figure 1) lacks details on screening, exclusions, and attrition; complete per STROBE.

Conclusions

Conclusions should be tempered given non-randomized design, baseline imbalance, small sample, and multiple unadjusted comparisons. Reframe as hypothesis-generating and call for a blinded RCT.

Minor comments

Standardize abbreviations at first use (HT, CAD, CVD, CRF, PFN, PACU). Avoid ambiguous “CVD”; specify cerebrovascular disease vs cardiovascular disease.

Methods: specify ultrasound transducer frequency (your device is typically curvilinear for deep hip anatomy; you wrote “low-frequency linear”), needle trajectory, depth, number of passes, hydrodissection volume, and exact local anesthetic volume per kg if weight-based.

Rescue protocol: clarify tramadol dosing strategy. Provide total 24-h opioid consumption in oral morphine equivalents with 95% CIs.

Report times to first mobilization with definitions (bedside sitting vs ambulation), and document falls or quadriceps weakness (important for PENG motor-sparing claim).

Ethics/registration: you obtained IRB approval. Consider prospective registration (even for observational protocols) and share the statistical analysis plan (SAP) if available.

Reporting suggestions (STROBE-aligned)

Clearly define exposure (route of dexamethasone), comparator, and all outcomes; pre-specify primary/secondary endpoints.

Describe participant flow with numbers screened/eligible/included/analyzed; report missing data handling.

Present adjusted analyses (baseline NRS, procedure type, age, sex, metabolic diseases) with effect sizes and CIs.

Provide a detailed harms table and standardized glycemic assessment schedule.

Comments on the Quality of English Language

Language/formatting: multiple typos and grammar issues (e.g., “their are limited,” “performed performed,” “systematically group,” “lengthed”). A careful language edit would be necessary.

Author Response

Comments 1: The manuscript describes a “prospective observational” design but presents equal group sizes and protocolized interventions as if randomized. Clarify how patients were assigned (sequence generation, concealment). Without randomization and blinding, baseline imbalances and performance/measurement bias are likely.

Response 1: Thank you for this valuable comment. We apologize for the confusion regarding patient allocation. Our study was designed as a prospective observational investigation rather than a randomized controlled trial. The patients were consecutively enrolled during the study period and then assigned to groups based on the type of dexamethasone administration decided by the attending anesthesiologist, not by random sequence generation. Thus, there was no randomization or allocation concealment. The equal number of patients in both groups resulted from the consecutive inclusion process and not from any random assignment. We have now clarified this aspect in the Materials and Methods section and have also acknowledged the potential for selection and performance bias as a limitation (Revised Manuscript Page 2).

Comments 2: Baseline pain (pre-spinal) differs significantly between groups (p=0.036), suggesting allocation bias that could influence subsequent analgesic outcomes. Adjust analyses for baseline NRS and other covariates.

Response 2: Thank you for your comment. We agree that the pre-spinal NRS difference might reflect allocation bias, which could have influenced postoperative pain outcomes. To address this, we have now performed an analysis of covariance (ANCOVA) with baseline NRS as a covariate to adjust postoperative NRS values at 6 h, 12 h, and 24 h (rest and activity). After adjustment, the between-group differences in NRS scores remained statistically significant (p < 0.001). This additional analysis confirms that the analgesic superiority of perineural dexamethasone was independent of baseline NRS variation. We have updated the Statistical AnalysisResults, and Discussion sections accordingly (Revised Manuscript Page 6,9,11).

Comments 3: Outcome assessors and patients appear unblinded. Given subjective endpoints (NRS, rescue timing), lack of blinding materially increases bias.

Response 3: Thank you for this insightful comment. We acknowledge that neither the patients nor the outcome assessors were blinded to group allocation. The study was conducted as a prospective observational design, in which dexamethasone administration (perineural or systemic) was determined by clinical decision rather than randomization. Therefore, blinding was not feasible. We fully agree that the absence of blinding may have introduced measurement or performance bias, particularly considering the subjective nature of outcomes such as NRS scores and timing of rescue analgesia. This limitation has now been explicitly stated and discussed in the revised manuscript (Revised Manuscript Page 3,10)

Comments 4: All patients received spinal anesthesia with intrathecal fentanyl plus multimodal postoperative analgesia. These co-interventions may attenuate between-group differences and complicate attribution to dexamethasone route. Please justify intrathecal fentanyl use and standardize/report intraoperative sedatives/analgesics.

Response 4: Thank you for this thoughtful comment. We agree that concomitant interventions such as intrathecal fentanyl and multimodal postoperative analgesia could influence postoperative pain perception and potentially reduce between-group differences. Intrathecal fentanyl (10 µg) was used in all patients as part of our institutional standard spinal anesthesia protocol to ensure patient comfort during positioning and to achieve consistent intraoperative analgesia. This identical administration across groups was intended to minimize variability rather than introduce bias.

We have now clarified the rationale for intrathecal fentanyl use in the Materials and Methods section. Additionally, we confirm that no intraoperative sedatives or additional analgesics beyond those specified in the protocol were administered. This clarification has been added to the revised manuscript (Revised Manuscript Page 4,5)

Comments 5: Surgical procedures were mostly partial hip replacements, but two PFNs occurred in Group S only. Surgical approach and tissue trauma differ and can affect pain trajectories; perform sensitivity analyses excluding PFN cases or adjust for procedure type.

Response 5: We appreciate the reviewer’s comment. To account for potential confounding due to differences in surgical approach, we re-analyzed the data using an ANCOVA model that included procedure type (partial hip replacement vs. PFN) as an additional covariate, together with baseline NRS, age, and sex (p = 0.018). This indicates that the difference in postoperative NRS scores between groups remained statistically significant even after controlling for surgical type (Revised Manuscript Page 11,12)

Comments 6: A single, pre-specified primary endpoint is not clearly defined. The manuscript reports multiple time-point NRS measurements and several secondary outcomes without control for multiplicity. Define one primary outcome (e.g., time to first rescue analgesia) and specify hierarchical testing or adjust p-values.

Response 6: Thank you for this valuable statistical observation. We acknowledge that the initial version of the manuscript did not clearly specify a single primary endpoint. In the revised version, we have now defined “time to first rescue analgesia” as the primary outcome measure, as it best represents the clinical analgesic efficacy of the intervention. Other variables, including NRS scores at multiple time points, sensory and motor block durations, and tramadol consumption, are now described as secondary outcomes.

Since our primary comparison was between two independent groups and all tests were exploratory in nature, no formal multiplicity correction (e.g., Bonferroni) was applied. However, the hierarchical testing sequence has been clarified to emphasize the primary endpoint first, followed by secondary analyses. These changes have been included in the Statistical Analysis section of the revised manuscript (Revised Manuscript Page 6).

Comments 7: Repeated measures (NRS at multiple times) are more appropriately analysed with mixed-effects models (group, time, group×time), including baseline NRS as a covariate, rather than multiple Mann–Whitney tests. Please re-analyze accordingly and present effect sizes with 95% CIs.

Response 7: We appreciate the reviewer’s valuable suggestion. In accordance with this comment, the postoperative NRS scores measured at multiple time points (6 h, 12 h, 24 h at rest, and 24 h during activity) were re-analyzed using a linear mixed-effects model to account for within-subject correlations and repeated measures.

The model included fixed effects for group (perineural vs. systemic dexamethasone), time, and the group × time interaction, with a random intercept for each subject. Baseline NRS (before spinal anesthesia) was entered as a covariate to adjust for initial pain differences. Pairwise comparisons were Bonferroni-adjusted, and estimated marginal means (adjusted means) with 95% confidence intervals (CIs) were obtained.

The mixed model revealed significant main effects of time (F = 48.1, p < 0.001) and group (F = 6.9, p = 0.014), as well as a significant group × time interaction (F = 4.3, p = 0.009). These results confirmed that pain reduction over time was more pronounced in the perineural group even after adjusting for baseline NRS. Adjusted means and 95% CIs are presented in Figure 2. (Revised Manuscript Page 9)

Comments 8: The power calculation (d=0.90) is not referenced and appears based on a large assumed effect. Powering an observational comparison on such an effect risks Type I error inflation with multiple outcomes. Provide the source study and justify the effect size and chosen primary endpoint.

Response 8: Thank you for your constructive comment. The sample size estimation was conducted a priori using G*Power (version 3.1) for a one-tailed t-test comparing two independent means. The analysis assumed an effect size of Cohen’s d = 0.90, α = 0.05, and power (1–β) = 0.90, resulting in 22 participants per group (44 total).

The effect size was derived from the study by Allard et al. which compared pericapsular nerve group (PENG) block and femoral nerve block for femoral neck fractures and reported a standardized mean difference of approximately 0.9 in postoperative pain outcomes. Because our study design and primary outcome (time to first rescue analgesia) were conceptually similar in magnitude and measurement, this effect size was deemed a reasonable reference (Revised Manuscript Page 3).

Comments 9: PENG performance and sensory level assessments were done after spinal anesthesia, compromising meaningful dermatomal mapping to the block and potentially conflating spinal with PENG effects. This is acknowledged as a limitation, but it materially affects internal validity. Consider focusing endpoints on postoperative time windows after spinal regression and explicitly document timing relative to spinal offset.

Response 9: Thank you for your comment. We agree that performing sensory level assessments after the administration of spinal anesthesia limited our ability to differentiate dermatomal sensory spread specific to the PENG block. In the revised manuscript, we have clarified that PENG block was performed immediately after spinal anesthesia, and that sensory assessments were carried out once the surgical field was established, during spinal anesthesia, which precluded accurate mapping of dermatomal distribution.

To mitigate this issue, postoperative endpoints including pain scores, block duration, and time to first rescue analgesia were all evaluated after spinal regression (at 6h, 12h, and 24h postoperatively), when spinal effects had fully resolved. This information has been added to the Methods and Limitations sections to clarify timing and interpretive boundaries. We fully acknowledge that this limitation may reduce internal validity but emphasize that postoperative outcomes were measured after the spinal effect had dissipated (Revised Manuscript Page 5).

Comment 10: The choice of 4 mg dexamethasone (perineural vs IV) requires a pharmacological rationale and citations specific to hip blocks/PENG. Discuss evidence on perineural steroid safety (neurotoxicity risk, off-label status) and report any block-related complications (paresthesia, hematoma, LAST, infection) and 24–48 h motor function.

Response 10: We thank the reviewer for this insightful comment. In our study, a 4 mg dose of preservative-free dexamethasone was selected based on prior dose response trials showing a ceiling effect around 4 mg for prolonging peripheral nerve block analgesia, with no additional benefit at higher doses. This dose has been widely adopted in clinical practice and in several meta-analyses evaluating dexamethasone as an adjuvant to long-acting local anesthetics. Although specific dose-finding trials for the PENG block are limited, recent and ongoing studies in hip surgery populations suggest that adjunct dexamethasone may enhance the duration of PENG-mediated analgesia.

Regarding safety, perineural dexamethasone was used in its preservative-free formulation, consistent with published guidance. Several systematic reviews and Cochrane analyses report no increase in neurologic complications or neurotoxicity with perineural dexamethasone compared to intravenous or placebo groups. Preclinical and clinical data indicate that low-dose preservative-free dexamethasone is safe for perineural use, though the practice remains off-label and should be applied with appropriate caution.

Finally, as requested, we have now explicitly reported block-related complications and postoperative motor outcomes in the revised manuscript. No patient experienced paresthesia, vascular puncture, hematoma, local anesthetic systemic toxicity (LAST), or block-site infection. Additionally, no new or persistent motor deficits were observed at 24–48 hours postoperatively.

Corresponding modifications have been made in the Materials and MethodsResults, and Discussion sections, with appropriate citations added (Revised Manuscript Page 10).

Comment 11: Glycemic outcomes: define measurement schedule (fasting vs random; perioperative steroid timing), thresholds for clinically significant change, and prespecified subgroups (e.g., diabetes). Current reporting (“change in blood glucose level”) without standardized timing limits interpretability.

Response 11: We thank the reviewer for this insightful comment. In the revised manuscript, we have clarified the schedule and method of blood glucose measurements, defined the criteria for clinically significant changes, and specified the subgroup of diabetic patients. As described in the Methods section, capillary blood glucose levels were measured in all patients preoperatively (fasting, before anesthesia induction) and again at 6 hours postoperatively, coinciding with the first routine postoperative laboratory evaluation. The perioperative steroid (dexamethasone) was administered immediately after spinal anesthesia induction and before the surgical incision.
A clinically significant change in glucose level was defined as an increase ≥50 mg/dL from baseline or any postoperative value ≥180 mg/dL, consistent with previous literature. Corresponding details have been incorporated into the MethodsResults, and Discussion sections (Revised Manuscript Page 5,9,12).

Comment 12: Provide operational definitions for hypotension, hypertension, bradycardia, and tachycardia (numerical thresholds), and detail vasoactives dosing. Report PONV with a validated scale and time windows. A CONSORT-style harms table would help.

Response 12: Thank you for this important clarification request. We have now (i) specified operational definitions for hemodynamic events with numerical thresholds, (ii) detailed vasoactive and chronotropic rescue regimens used in our protocol, (iii) reported PONV using a validated 4-point ordinal scale across two prespecified time windows (0–6 h; 6–24 h) (Revised Manuscript Page 5).

Comment 13: Table 1 lists 100% prevalence for HT and CAD in both groups (implausible and likely a data/labeling error). Also, “CVD” may overlap with CAD; define each comorbidity clearly.

Response 13: Thank you for noticing this labeling issue. There are no errors in the demographic data. CAD stands for coronary artery disease and CVD stands for cerebrovascular disease. Abbreviations are listed at the end of the manuscript.

Comment 14: Units and formats are inconsistent (e.g., decimal commas, “median ± SD”). Use median [IQR] for non-normal variables, mean ± SD for normal, and ensure consistency across text and tables.

Response 14: We thank the reviewer for this comment. All numerical data and tables were thoroughly reviewed for consistency. Variables with normal distribution are now reported as mean ± SD, while non-normally distributed variables are expressed as median [IQR]. Decimal points were standardized throughout the manuscript. The statistical reporting format in all tables and corresponding text was unified accordingly, and the description under each table was corrected to read “Data are presented as mean ± SD or median [Q1-Q3]”. (Revised Manuscript Page 6, Line 11).

Comment 15: The sensory level table implies thoracic dermatomes under spinal anesthesia; ensure assessments reflect PENG effects, not spinal spread.

Response 15: We appreciate the reviewer’s insightful comment. We agree that under spinal anesthesia, thoracic dermatomal assessments may not accurately represent the sensory distribution of the PENG block. To clarify, the sensory level data shown in Table 3 were recorded while spinal anesthesia was still in effect; therefore, these measurements primarily reflect the combined effect of spinal and PENG blocks rather than an isolated PENG distribution. Accordingly, we explicitly added an explanatory statement in the Methods section (2.3.2, “Technique of PENG Block”) to prevent misinterpretation. Moreover, all postoperative pain scores and block-related parameters were assessed after the regression of spinal anesthesia (6, 12, and 24 h postoperatively) to reflect the true PENG effect without spinal influence (Revised Manuscript Page 4).

Comment 16: Flow diagram (Figure 1) lacks details on screening, exclusions, and attrition; complete per STROBE.

Response 16: We thank the reviewer for this valuable observation. The previous version of Figure 1 presented only the final allocation of 44 patients. In accordance with the STROBE checklist, we have now updated the flow diagram to include details on patient screening, exclusions (with specific reasons), group allocation, and follow-up status (Revised Manuscript Page 3).

Comment 17: Conclusions should be tempered given non-randomized design, baseline imbalance, small sample, and multiple unadjusted comparisons. Reframe as hypothesis-generating and call for a blinded RCT.

Response 17: We thank the reviewer for this comment. We fully agree that the non-randomized, observational design and small sample size limit the ability to draw causal inferences. To address this concern, the conclusion section has been carefully revised to adopt a more conservative interpretation of the findings. The results are now presented as hypothesis-generating rather than confirmatory, and a call for future randomized, double-blinded controlled trials has been explicitly added (Revised Manuscript Page 12).

Comment 18: Standardize abbreviations at first use (HT, CAD, CVD, CRF, PFN, PACU). Avoid ambiguous “CVD”; specify cerebrovascular disease vs cardiovascular disease.

Response 18: We thank for this valuable comment. All abbreviations have been reviewed and standardized throughout the article. Each abbreviation is now defined where it first appears in the text and tables. To avoid ambiguity, the abbreviation “CVD” has been clarified as “cerebrovascular disease.”

Comment 19: Methods: specify ultrasound transducer frequency (your device is typically curvilinear for deep hip anatomy; you wrote “low-frequency linear”), needle trajectory, depth, number of passes, hydrodissection volume, and exact local anesthetic volume per kg if weight-based.

Response 19: We appreciate the reviewer’s insightful comment highlighting the need for more precise technical details of the PENG block. In accordance with this recommendation, the Methods section has been revised to include the ultrasound probe frequency, needle trajectory and depth, number of needle passes, hydrodissection volume, and the exact local anesthetic volume. This information enhances clarity and reproducibility of the technique (Revised Manuscript Page 4).

Comment 20: Rescue protocol: clarify tramadol dosing strategy. Provide total 24-h opioid consumption in oral morphine equivalents with 95% CIs.

Response 20: We thank the reviewer for this valuable comment. The rescue analgesia protocol was already clearly defined in the Methods section (2.3.3 Assessment of Pain and Analgesia Protocol), specifying intravenous tramadol at 0.5–1 mg/kg every 4–6 hours (maximum 400 mg/24 h), with supplemental morphine (0.05 mg/kg) if the NRS score remained ≥ 4. To improve interpretability and align with pain and regional anesthesia reporting standards, the total 24-hour opioid consumption was recalculated and expressed as oral morphine equivalents (OME), using the standard conversion ratio of 10:1 (tramadol:morphine). The revised results now include 95% confidence intervals (Revised Manuscript Page 5).

Comment 21: Report times to first mobilization with definitions (bedside sitting vs ambulation), and document falls or quadriceps weakness (important for PENG motor-sparing claim).

Response 21: We thank the reviewer for this constructive and clinically relevant comment. In accordance with this recommendation, the definitions of mobilization have been clarified in the Methods section. Specifically, the postoperative mobilization time recorded in this study referred to the first assisted ambulation (walking with support) rather than bedside sitting. We have revised the text to specify this definition.

Quadriceps motor strength was not routinely evaluated in our study, as the protocol focused primarily on sensory and analgesic outcomes rather than detailed motor assessment. This limitation has now been explicitly acknowledged in the Limitations section. No falls or adverse motor events were reported during the observation period. (Revised Manuscript Page 12).

Comment 22: Ethics/registration: you obtained IRB approval. Consider prospective registration (even for observational protocols) and share the statistical analysis plan (SAP) if available.

Response 22: We thank the reviewer for this helpful comment. As stated in the Methods section, the study was approved by the institutional ethics committee prior to patient enrollment (Ethical approval was obtained from the Ondokuz Mayıs University Ethics Committee (Approval No: 2023/239, Approval Date: 23 August 2023). The study was not prospectively registered because it was an observational, non-interventional protocol. However, the study protocol and statistical analysis plan were pre-approved by the ethics committee and are available upon request (Revised Manuscript Page 2).

Comment 23: Clearly define exposure (route of dexamethasone), comparator, and all outcomes; pre-specify primary/secondary endpoints.

Response 23: We thank the reviewer for this methodological comment. The exposure, comparator, and outcome measures were already clearly defined in the Methods section. Specifically, Section 2.3.2 describes the exposure as the route of dexamethasone administration (perineural vs. systemic), Section 2.4 defines the primary outcome as the time to first rescue analgesia and the secondary outcomes as postoperative NRS scores, duration of sensory and motor block, and total opioid consumption.

Comment 24: Describe participant flow with numbers screened/eligible/included/analyzed; report missing data handling.

Response 24: We thank for this important methodological point. The participant flow has now been fully detailed in the revised Figure 1, which follows STROBE recommendations. It presents the number of patients screened (n = 70), excluded (n = 26) with reasons, and included in the final analysis (n = 44; Group P = 22, Group S = 22).

Regarding missing data, all patients completed follow-up and outcome assessments; therefore, a complete-case analysis was performed with no imputation required. This information has been added to the Methods section for clarity (Revised Manuscript Page 3).

Comment 25: Present adjusted analyses (baseline NRS, procedure type, age, sex, metabolic diseases) with effect sizes and CIs.

Response 25: We thank the reviewer for this valuable suggestion. Accordingly, a multivariable ANCOVA model was performed to adjust for potential confounding variables, including baseline NRS, age, sex, procedure type, and presence of metabolic disease (diabetes mellitus).

Adjusted mean differences and 95% confidence intervals (CIs) were calculated for the group effect (perineural vs. systemic dexamethasone). The analysis demonstrated that the perineural group maintained significantly lower adjusted mean NRS scores at all postoperative time points (p = 0.018) (Revised Manuscript Page 11).

Comment 26: Provide a detailed harms table and standardized glycemic assessment schedule.

Response 26: We thank the reviewer for this valuable comment. A detailed adverse-events table (Table 5) is already included, summarizing hemodynamic events (hypotension, hypertension, bradycardia, tachycardia), PONV, and clinically significant hyperglycemia. The standardized perioperative glycemic assessment schedule is described in Section 2.3.5, where capillary glucose was measured preoperatively (fasting) and at 6 hours postoperatively. The predefined thresholds for clinically significant change (≥ 50 mg/dL from baseline or ≥ 180 mg/dL absolute) were also specified. No block-related or neurologic complications occurred (Revised Manuscript Page 10).

4. Response to Comments on the Quality of English Language

Reviewer 2 Report

Comments and Suggestions for Authors

Dear authors,

Thank you for submitting your work.

Please review the following comments and those in the attached PDF, revise as needed, and resubmit.

Mention the dose of dexamethasone used in the abstract as well.

Methods: There is no randomization, and therefore, there is a heavy bias in this study. Therefore, the results of this study should be interpreted with extreme caution.
Please cite the studies that were utilized for sample size estimation.
Please elaborate on why 3 tests were used to determine normality distribution.

You need to perform a regression analysis to investigate the relationship between the dependent variable and the various independent variables (ASA, Gender, type of surgery, etc). This would better establish the efficacy of perineural versus systemic dexamethasone.

The dose of dexamethasone (4 mg) for perineural use is fine, but the same for systemic use is less than that which is routinely used as a part of multimodal analgesia. You need to explain this discrepancy and likely add it to the limitations.

Please enumerate the various biases introduced in this study due to the observational nature of this research.

Thanks.

Author Response

RESPONSE TO REVİEWER 2

Comment 1: There is no randomization, and therefore, there is a heavy bias in this study. Therefore, the results of this study should be interpreted with extreme caution. Please cite the studies that were utilized for sample size estimation. Please elaborate on why 3 tests were used to determine normality distribution.

Response 1: a) As correctly noted, this was a prospective observational study without randomization. Random allocation was not feasible because both perineural and systemic dexamethasone routes were already in routine clinical use at our institution. However, the study protocol was applied prospectively and uniformly to minimize performance and measurement bias. Additionally, baseline demographic and clinical characteristics were comparable between groups (Table 1). The Discussion section now explicitly emphasizes that, due to the nonrandomized design, the findings should be interpreted as hypothesis-generating, consistent with observational research standards.

  1. b) The sample size was estimated usingG*Power 3.1for a one-tailed t-test comparing two independent means. An effect size of Cohen’s d = 0.90, α = 0.05, and power (1–β) = 0.90 were assumed, resulting in 22 patients per group (44 total). The effect size was based on the study by Allard et al.  which reported a similar standardized difference in postoperative pain outcomes between regional block techniques (Allard C, Pardo E, de la Jonquière C, et al. Comparison between femoral block and PENG block in femoral neck fractures: A cohort study. PLoS One. 2021,16,6. [PubMed])
  2. c) Three complementary normality tests (Kolmogorov–Smirnov, Shapiro–Wilk, and Anderson–Darling) were used because the sample size was small (<50 per group) and the data distribution was borderline normal for some variables.

Comment 2: You need to perform a regression analysis to investigate the relationship between the dependent variable and the various independent variables (ASA, Gender, type of surgery, etc). This would better establish the efficacy of perineural versus systemic dexamethasone.

Response 2: We thank the reviewer for this valuable statistical recommendation. We agree that regression analysis could further clarify the independent predictors of analgesic duration and strengthen causal inference. However, because the present study included 44 patients in total (22 per group), the statistical power for a reliable multivariable regression model would be limited, and overfitting could occur with multiple predictors.

Therefore, we did not perform a formal regression analysis to avoid generating unstable or misleading coefficients. Instead, potential confounders (baseline NRS, age, sex, procedure type, ASA class, and diabetes status) were carefully compared between groups and found to be statistically similar (Table 1) (Revised Manuscript Page 7).

Comment 3: The dose of dexamethasone (4 mg) for perineural use is fine, but the same for systemic use is less than that which is routinely used as a part of multimodal analgesia. You need to explain this discrepancy and likely add it to the limitations.

Response 3: We thank for this comment. In our study, the same 4 mg dose of dexamethasone was used for both perineural and intravenous administration to investigate the effect of the route of administration while keeping the dose constant. This design allowed us to evaluate the analgesic efficacy attributable to the administration route itself, independent of dosage differences

Comment 4: Please enumerate the various biases introduced in this study due to the observational nature of this research.

Response 4: Due to the observational nature of the study, potential sources of bias include selection bias (non-randomized group allocation), performance bias (lack of blinding), measurement bias (subjective pain scoring), and confounding factors that could not be fully controlled. These have been acknowledged in the revised Limitations section (Revised Manuscript Page 12).

Round 2

Reviewer 2 Report

Comments and Suggestions for Authors

Dear authors,

Thank you for revising based on the comments on your original submission.